# Causally Robust Reward Learning from Reason-Augmented Preference Feedback

**Minjune Hwang**[1*]     **Yigit Korkmaz**[1]     **Daniel Seita**[1†]     **Erdem Bıyık**[1†]

[1]Thomas Lord Department of Computer Science, University of Southern California
[*]Corresponding author     [†]Equal advising
`{minjuneh,ykorkmaz,seita,biyik}@usc.edu`

## Abstract

Preference-based reward learning is widely used for shaping agent behavior to match a user's preference, yet its sparse binary feedback makes it especially vulnerable to causal confusion. The learned reward often latches onto spurious features that merely co-occur with preferred trajectories during training, collapsing when those correlations disappear or reverse at test time. We introduce ReCouPLe, a lightweight framework that uses natural language rationales to provide the missing causal signal. Each rationale is treated as a guiding projection axis in an embedding space, training the model to score trajectories based on features aligned with that axis while de-emphasizing context that is unrelated to the stated reason. Because the same rationales (e.g., "*avoids collisions*", "*completes the task faster*") can appear across multiple tasks, ReCouPLe naturally reuses the same causal direction whenever tasks share semantics, and transfers preference knowledge to novel tasks without extra data or language-model fine-tuning. Our learned reward model can ground preferences on the articulated reason, aligning better with user intent and generalizing beyond spurious features. ReCouPLe outperforms baselines by up to 1.5x in reward accuracy under distribution shifts, and 2x in downstream policy performance in novel tasks. We have released our code at `https://github.com/mj-hwang/ReCouPLe`.

## 1 Introduction

Designing reward functions that faithfully capture human intent is one of the central obstacles to deploying learning agents in the real world. Preference-based reinforcement learning (PbRL) removes the need for hand-crafted rewards by asking a human to compare two trajectories and indicate their preference (Christiano et al., 2017; Sadigh et al., 2017; Bıyık et al., 2019; Hejna & Sadigh, 2023; Lee et al., 2021; Ouyang et al., 2022). Unfortunately, this binary feedback conveys at most a single bit of information and leaves the reward model free to explain the preference with any correlating feature in its observation space. Under the presence of non-causal distractor features that are spuriously correlated with preference labels, reward models often learn to rely on such features (Tien et al., 2023). These features, however, are irrelevant to the task success. When they disappear or change at test time, the agent can suffer from reward misidentification and fail to generalize. Since each comparison supplies so little information, it leaves many causal explanations indistinguishable. Without extra guidance, the learner cannot tell whether users prefer a trajectory for its smoothness, its speed, or some spurious cue in the background.

For example, suppose we want to train a robotic arm to pick up a box large enough to store toys (Figure 1). During data collection, every preference query shows a large red box and a small blue box, and the annotator always prefers the former. Because size and color are perfectly correlated in these comparisons, a reward model can reach zero training error by attending to the color cue instead of true size. At test time, when it encounters a large blue box next to a small red box, the learned reward could mistakenly favor the small red box.

A natural way to overcome this ambiguity is to supply richer feedback. Following advancements in natural language processing, recent works in robot learning employed language for task planning (Ahn et al., 2022; Sharma et al., 2022; Singh et al., 2023; Bucker et al., 2022), policy learning (Cui

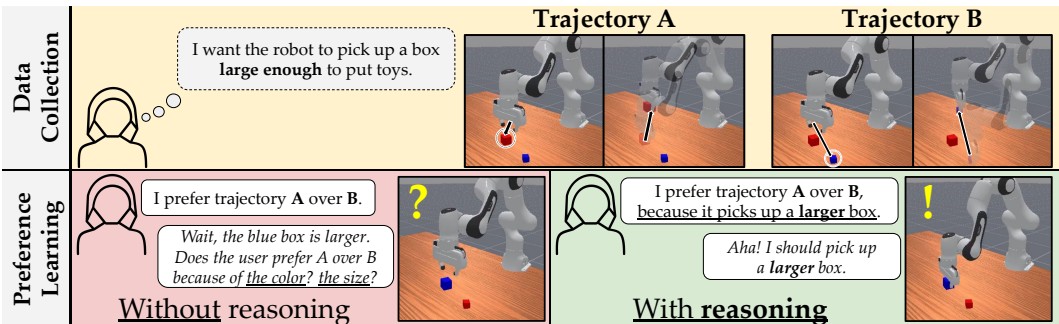

Figure 1: Preference learning can be susceptible to causal confusion, especially with the presence of non-causal distractor features that merely co-occur with preferred trajectories. In the example above, the reward model struggles to identify the exact feature of a trajectory that determined the user's preference. By providing a reason, the agent can identify the causal feature.

et al., 2023; Dai et al., 2025; Shi et al., 2024), and reward shaping (Goyal et al., 2019; Yang et al., 2024; Holk et al., 2024). We claim that short natural language *rationales* carry exactly the causal signal the model is missing. "*I prefer this trajectory because it picks up the large box*" tells the learner which feature matters for the user's preference.

We present **ReCouPLe** (**Re**ason-based **Co**n**fu**sion Mitigation in **P**reference **Le**arning), a lightweight framework which consists of a data modality that couples preferences with their reasons/rationales and a learning algorithm that treats each rationale as a directional guide in a shared trajectory-language representation space. We design a simple loss that encourages the preference to be based on the direction of the reason specified by the user, rather than on incidental correlations in a pair of trajectories. The language encoder shared across all tasks makes sure we retain the same semantics between them. Subsequently, this decoupling of reason components enables us to leverage shared rationales appearing across tasks, learning a reward function that generalizes zero-shot from one task to another.

In summary, our contributions are:

1. Based on the observation that pairwise preferences provide limited information in the presence of non-causal distractor features, leaving reward models prone to causal confusion, we design a new feedback type by supplying complementary causal cues that help disambiguate the true preference signal.
2. We introduce ReCouPLe, a framework that injects causal structure into preference learning by aligning trajectory representations with rationale embeddings.
3. We demonstrate that augmenting comparisons with rationales produces reward models that significantly reduce causal confusion relative to state-of-the-art baselines. Moreover, these models transfer across tasks without additional preference queries by leveraging shared underlying reasons.

## 2 RELATED WORK

**Learning from Human Feedback.** Policy learning from human feedback has taken many forms, including demonstrations (Schaal, 1996; Ho & Ermon, 2016), corrections (Bajcsy et al., 2017; 2018), interventions (Korkmaz & Bıyık, 2025; Kelly et al., 2019), language (Shi et al., 2024; Sharma et al., 2022), and preferences (Bıyık et al., 2019; Christiano et al., 2017; Hejna & Sadigh, 2023). Among these, preferences and language occupy opposite ends of the spectrum: binary comparisons are intuitive and easy to provide but limited in expressivity, whereas natural language is rich and flexible but often underconstrained due to the vagueness and ambiguity of natural language, necessitating an additional modality (Casper et al., 2023; Zeng et al., 2023; Shi et al., 2024; Cui et al., 2023). In this work, we complement pairwise comparisons with natural language rationales, which provides richer causal information while preserving the ease of use.

**Preference-based Learning.** Preference-based methods have become popular for reinforcement learning, especially where explicit rewards are unavailable (Christiano et al., 2017; Sadigh et al., 2017; Hejna & Sadigh, 2023; Bıyık et al., 2019). While effective, their reliance on binary comparisons poses two main challenges: (i) each query conveys at most one bit of information, and (ii) the feedback

is ambiguous when multiple task-relevant features exist (Casper et al., 2023). This makes reward models prone to relying on spurious, non-causal cues (Tien et al., 2023). When policies are optimized against such flawed proxies, this vulnerability frequently triggers "Causal Goodhart" effects, where optimizing a learned proxy reward degrades actual policy performance because the proxy relies on features that correlate with, but do not cause, the desired behavior (Manheim & Garrabrant, 2018; Gao et al., 2023). Prior efforts to enrich preference signals include active query selection strategies such as information gain (Bıyık et al., 2019), volume removal (Sadigh et al., 2017) and maximum regret (Wilde et al., 2020), and augmentations with feature-level queries (Basu et al., 2018). More recently, Peng et al. (2024) proposed feature-wise preference learning, asking users *why an example is preferred*. However, these methods assume access to structured, task-specific features and have been demonstrated in limited settings like linear bandits. In contrast, we employ free-form natural language rationales alongside binary preferences, aligning the reward model with causal explanations while reducing reliance on spurious correlations.

**Robot Learning with Language Feedback.** There have been several recent works that utilize natural language for improving learned robot policies via different strategies such as task planning (Ahn et al., 2022; Singh et al., 2023), policy learning (Cui et al., 2023; Dai et al., 2025; Shi et al., 2024), and reward shaping (Goyal et al., 2019). Shi et al. (2024) employ language-conditioned behavior cloning (LCBC) for corrective language commands and improving policies. Cui et al. (2023) introduce an approach to use human language feedback to correct robot manipulation in real-time via shared autonomy. Dai et al. (2025) propose a data generation pipeline that automatically augments expert demonstrations with failure recovery trajectories and fine-grained language annotations for training recovery policies. In the domain of preference learning, Yang et al. (2024) learn a shared latent space for trajectories and comparative language like "move farther from the stove", showing that language can make reward learning faster and more intuitive. While most existing methods treat language merely as an additional conditioning input, Holk et al. (2024) explicitly leverage natural language reasoning to mitigate causal confusion. However, like the feature-level preferences of Peng et al. (2024), their approach relies on a predefined set of intrinsic features. Specifically, they prompt an LLM to analyze human text prompts and extract the sentiment and magnitude associated with each predefined feature to guide reward modeling. In contrast, our approach directly uses language embeddings of free-form rationales in a shared trajectory-language space, removing the bottleneck of predefined features and enabling better transfer across tasks.

## 3 PRELIMINARIES

We consider a collection of tasks modeled as finite-horizon Markov decision processes (MDPs) $\mathcal{M} = (\mathcal{S}, \mathcal{A}, P, r, \gamma, T)$, where $\mathcal{S}$ is the state space, $\mathcal{A}$ is the action space, $P(s_{t+1} \mid s_t, a_t)$ is the transition kernel, $r : \mathcal{S} \times \mathcal{A} \to \mathbb{R}$ is the reward function, $\gamma \in [0, 1)$ is the discount factor, and $T$ is the maximum time horizon. The reward function $r$ is unknown and must be inferred from the user's pairwise preference feedback augmented with rationales.

### 3.1 REWARD LEARNING FROM PREFERENCE DATA

We assume access to preference data in the form of binary comparisons. Given a pair of trajectory segments $(\tau_A, \tau_B)$ of horizon $H \leq T$, a user (either human, or a proxy) provides preference label $y$:

$$y = \begin{cases} 1 & \text{if } \tau_A \succ \tau_B, \\ 0 & \text{otherwise.} \end{cases}$$

where each segment is a sequence of observations and actions $(s_k, a_k, \ldots, s_{k+H}, a_{k+H})$. Following the Bradley-Terry model (Bradley & Terry, 1952), the probability that the trajectory $\tau_A$ is preferred over the trajectory $\tau_B$ is given by:

$$P_r(\tau_A \succ \tau_B) = \frac{\exp(r(\tau_A))}{\exp(r(\tau_A)) + \exp(r(\tau_B))} \tag{1}$$

where $r(\tau)$ is the cumulative discounted reward of trajectory $\tau$. In order to infer the reward function, prior works (Christiano et al., 2017; Sadigh et al., 2017; Lee et al., 2021) in preference-based RL train a reward function $\hat{r}_\omega : \mathcal{S} \times \mathcal{A} \to \mathbb{R}$ parameterized by $\omega$, on a dataset $D$ of $(\tau_A, \tau_B, y)$ triplets by minimizing the binary cross-entropy (BCE) loss with the Bradley-Terry model:

$$\mathcal{L}_{\text{BCE-BT}} = -\mathbb{E}_{(\tau_A, \tau_B, y) \sim D}[y \log P_{\hat{r}_\omega}(\tau_A \succ \tau_B) + (1 - y) \log(1 - P_{\hat{r}_\omega}(\tau_A \succ \tau_B))] \tag{2}$$

## 3.2 LANGUAGE INTERFACES

In addition to the standard MDP formulation, each task is associated with a **task description** $\ell_{\text{task}}$, a short instruction such as "pick up the cup" or "push the cube." These descriptions provide high-level semantic grounding for the task but are not sufficient on their own to fully specify the reward function. To capture the finer distinctions that matter to the users, we rely on preference queries. Additionally, each preference label has an optional **reason** $\ell_{\text{reason}}$ that explains why one trajectory is preferred over the other (e.g., "because it avoids collisions"). A frozen language encoder LM maps these strings to fixed embeddings of dimensionality $d$: $\theta = \text{LM}(\ell_{\text{task}}) \in \mathbb{R}^d$ and $\psi = \text{LM}(\ell_{\text{reason}}) \in \mathbb{R}^d$.

## 4 METHOD

Preference-based reinforcement learning typically fits a single-task reward by maximizing the likelihood of observed comparisons (Eq. 1). We extend this framework to the multi-task setting, where each task is identified by its language description. Specifically, we model the reward as the inner product between the trajectory representation and the task embedding:

$$r(\tau, \ell_{\text{task}}) \ = \ \phi(\tau)^\top \text{LM}(\ell_{\text{task}}) = \ \phi(\tau)^\top \theta, \tag{3}$$

where the trajectory encoder $\phi : \tau \to \mathbb{R}^d$ is the only trainable component, as the task embedding $\theta = \text{LM}(\ell_{\text{task}})$ is frozen. We use this reward formulation across all methods for consistency. Although linear in structure, the nonlinearity of both the trainable trajectory encoder and the frozen language model allows this simple form to capture complex, task-specific reward structures, similar to CLIP-based reward models (Sontakke et al., 2023). In our experiments, we use the pretrained T5 (Raffel et al., 2020) language model encoder as LM.

### 4.1 RECOUPLE - REASON-BASED CONFUSION MITIGATION IN PREFERENCE LEARNING

Our key idea is that short natural language rationales may reveal the causal features underlying a preference and thereby mitigate spurious correlations. For example, the statement *"I prefer this path because it avoids collisions"* explicitly identifies the relevant factor influencing the user's choice. Given a rationale $\ell_{\text{reason}}$, we obtain its embedding via the frozen language encoder, $\psi = \text{LM}(\ell_{\text{reason}})$. ReCouPLe then treats $\psi$ as a projection axis and decomposes the trajectory embedding $\phi(\tau)$ into two components: (i) a reason-aligned part parallel to $\psi$, and (ii) an orthogonal part corresponding to features unrelated to the rationale. Figure 2 visualizes this process.

Mathematically, this orthogonal projection onto $\psi$ induces two disjoint subspaces:

$$\phi(\tau) = \underbrace{\phi_{\|}(\tau)}_{\text{reason-aligned}} + \underbrace{\phi_{\perp}(\tau)}_{\text{reason-orthogonal}} \ , \quad \phi_{\|}^\top \phi_{\perp} = 0,$$

which is achieved by

$$\phi_{\|}(\tau) = \Big( \frac{\phi(\tau)^\top \psi}{\|\psi\|_2^2} \Big) \psi, \quad \phi_{\perp}(\tau) = \phi(\tau) - \phi_{\|}(\tau)$$

Subsequently, the reward term (Eq. 3) decomposes as

$$r(\tau, \ell_{\text{task}}) = \underbrace{r_{\|}(\tau, \ell_{\text{task}})}_{\text{explained by rationale}} + \underbrace{r_{\perp}(\tau, \ell_{\text{task}})}_{\text{residual task signal}} \ ,$$

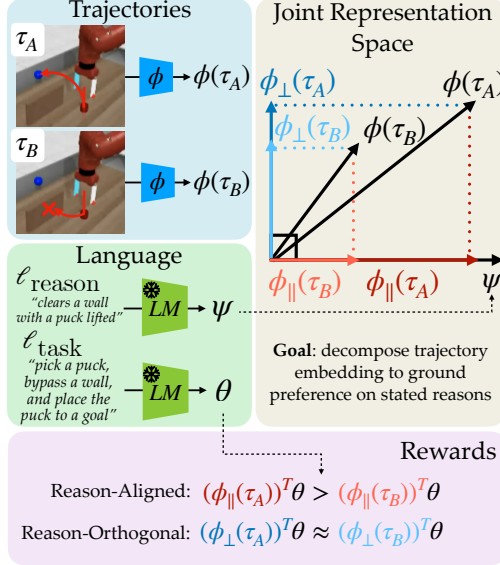

Reason-Aligned: $(\phi_{\|}(\tau_A))^T \theta > (\phi_{\|}(\tau_B))^T \theta$

Reason-Orthogonal: $(\phi_{\perp}(\tau_A))^T \theta \approx (\phi_{\perp}(\tau_B))^T \theta$

Figure 2: ReCouPLe decomposes the task reward by orthogonally projecting the trajectory representation to the reason language embedding and decomposing the representation into reason-aligned and reason-orthogonal components. This allows the reward model to isolate the causal feature specified in the rationale to explain the user's preference.

with the following components:

- $r_{\parallel}(\tau, \ell_{\text{task}}) = \phi_{\parallel}^{\top} \theta$ is the reason-aligned, *causal* component explicitly justified by the user's rationale.
- $r_{\perp}(\tau, \ell_{\text{task}}) = \phi_{\perp}^{\top} \theta$ is the *orthogonal* component that captures any task-relevant information the rationale overlooks (e.g., shaping rewards or domain priors).

Our insight is to ground the pairwise preference on the stated reason and prevent the model from relying on incidental correlations. This is achieved by forcing reward differences to depend solely on $r_{\parallel}$ while holding $r_{\perp}$ neutral.

Given the decomposed reward, we train our trajectory representation by three loss terms:

1. **Reason loss**: BCE loss using Bradley-Terry model with $r_{\parallel}$ *only*, enforcing that preferences are explained through the stated causal feature.

$$\mathcal{L}_{\text{reason}} = -\mathbb{E}_{(\tau_A, \tau_B, y) \sim D} [y \log P_{r_{\parallel}}(\tau_A \succ \tau_B) + (1 - y) \log(1 - P_{r_{\parallel}}(\tau_A \succ \tau_B))] \quad (4)$$

2. **Orthogonal consistency loss**: Consistency loss prevents non-causal features explaining pairwise preferences but still encourages them to capture any task-relevant information.
   We propose two different versions of this term, resulting in two variants of ReCouPLe:

   (a) **ReCouPLe-EC**: *Equality* constraint $r_{\perp}(\tau_A, \ell_{\text{task}}) \approx r_{\perp}(\tau_B, \ell_{\text{task}})$ for every comparison, ensuring $\phi_{\perp}$ carries no preference signal:

   $$\mathcal{L}_{\text{eq}} = \left( r_{\perp}(\tau_A, \ell_{\text{task}}) - r_{\perp}(\tau_B, \ell_{\text{task}}) \right)^2 \quad (5)$$

   (b) **ReCouPLe-IC**: *Inequality* constraint encouraging the difference between $r_{\parallel}$ to be greater than that of $r_{\perp}$ for a given pair of trajectories. A regularizer term computed by taking the BCE loss on total reward ($r_{\parallel} + r_{\perp}$) (Eq. 2) is also added to prevent potential mode collapses:

   $$\text{diff}_r(A \succ B) = r(\tau_A, \ell_{\text{task}}) - r(\tau_B, \ell_{\text{task}})$$

   $$S(A \succ B) = \frac{\exp(\text{diff}_{r_{\parallel}}(A \succ B))}{\exp(\text{diff}_{r_{\parallel}}(A \succ B)) + \exp(\text{diff}_{r_{\perp}}(A \succ B))}$$

   $$\mathcal{L}_{\text{ineq}} = -\mathbb{E}_{(\tau_A, \tau_B, y) \sim D} [y \log S(A \succ B) + (1 - y) \log(1 - S(A \succ B))] + \mathcal{L}_{\text{BCE-BT}} \quad (6)$$

   ReCouPLe-EC imposes a strict condition, requiring the reason-orthogonal components to be identical for compared trajectories. In contrast, ReCouPLe-IC is less restrictive, incentivizing differences in the reason-aligned component to dominate differences in total task rewards. As such, ReCouPLe-EC is more effective when a small, recurring set of reasons largely governs preferences and variability in the reason-orthogonal components is minimal, whereas ReCouPLe-IC is preferable when many plausible reasons may explain comparisons and when variation in reason-orthogonal reward is non-negligible.

3. **Reward-ratio regularizer**: $\mathcal{L}_{\text{ratio}}$ for keeping the magnitude of $r_{\parallel}$ below a fraction $\alpha$ of the total reward magnitude $r_{\parallel} + r_{\perp}$, preventing the trivial collapse of the reward into the causal subspace: $\mathcal{L}_{\text{ratio}} = \text{ReLU}\left( \frac{|r_{\parallel}|}{|r_{\parallel}| + |r_{\perp}| + \epsilon} - \alpha \right)$, where a small constant $\epsilon$ is included in the denominator to prevent division by zero.

The final objective is the following:

$$\mathcal{L}_{\text{ReCouPLe}} = \mathcal{L}_{\text{reason}} + \lambda_{\text{ratio}} \mathcal{L}_{\text{ratio}} + \begin{cases} \lambda_{\text{eq}} \mathcal{L}_{\text{eq}} & \text{(ReCouPLe-EC)}, \\ \lambda_{\text{ineq}} \mathcal{L}_{\text{ineq}} & \text{(ReCouPLe-IC)}. \end{cases}$$

## 5 EXPERIMENTS

We evaluate ReCouPLe on two complementary suites that probe distinct facets of the method. The first suite focuses on causal robustness in a single visuomotor task whose visual cues are deliberately confounded; the second investigates cross-task generalization in a multi-task manipulation benchmark. Together they address two research questions:

- **RQ1** (Robustness against causal confusion): Can ReCouPLe maintain preference accuracy when the covariate distribution shifts in a way that exposes spurious correlations?
- **RQ2** (Task transfer): Does the reason-aligned subspace learned on a small set of tasks transfer to a semantically related, novel task without additional preference queries?

We address **RQ1** with a custom ManiSkill (Tao et al., 2025) suite that deliberately entangles distractor features (object *color*) with the ground-truth causal factor (object *size*) and then evaluates under a *color-swapped* distribution. We assess **RQ2** with a set of Meta-World (Yu et al., 2020) tasks that are widely used to test few-shot/zero-shot transfer.

For both experiments, we compute a per-step embedding by first encoding each state-action pair $(s_t, a_t)$ with a modality-appropriate encoder: a convolutional encoder for image observations (Yarats et al., 2021) and a fully connected network for state-based control tasks. Concretely, we use a neural network encoder $e : \mathcal{S} \times \mathcal{A} \to \mathbb{R}^d$ to encode every state-action pair of a trajectory into the corresponding per-step embedding. We then obtain a trajectory embedding $\phi(\tau)$ by aggregating these stepwise embeddings over time. This additive design mirrors prior works in preference learning that formulates trajectory features as a sum of per-step features (Yang et al., 2024).

**Baselines.** We compare ReCouPLe against two baselines that share the same multi-task reward formulation but differ in the loss terms used to train the trajectory encoder $\phi$.

- **Multi-Task Bradley-Terry (BT-Multi)**: This baseline learns the trajectory encoder, $\phi$ by minimizing the binary cross-entropy loss $\mathcal{L}_{\text{BCE-BT}}$ (Eq. 2) across all tasks, using the shared reward definition above and ignoring the rationale $\ell_{\text{reason}}$. It therefore serves as the baseline without reason inputs.
- **Reason-Feature Preference (RFP)**: This baseline is an extension of *Pragmatic Feature Preferences* (PFP) (Peng et al., 2024). PFP assumes that each state can be represented by an *explicit, hand-designed feature vector*. Humans (i) specify which of those features are relevant to the task and (ii) give pairwise labels that compare *each relevant feature* across two items. In our setting, such features do not exist. Instead, we treat the frozen rationale embedding $\psi = \text{LM}(\ell_{\text{reason}})$ as a single *implicit* feature direction in the representation space. Besides the shared reward $r(\tau, \ell_{\text{task}})$ from Eq. 3, we also define a reason score, $q(\tau, \ell_{\text{reason}}) = \phi(\tau)^\top \psi$, and minimize the standard BCE loss from Eq. 2 and an additional auxiliary BCE loss term for the specified reason's score.

**Metrics.** After training each reward model, we report *reward accuracy*: the proportion of held-out preference pairs where the preferred trajectory is labeled with a higher reward. We evaluate ManiSkill on in-distribution (ID) and color-swapped (OOD; out-of-distribution) splits, and Meta-World on three training tasks (validation split) plus a held-out novel task. We then validate the reward model's usefulness in downstream policy learning. To assess this, we use the learned model to assign reward values to trajectories in offline play data and train an offline RL policy for each task to report its task success rate. These offline datasets have trajectories with different levels of optimality, so simply imitating the behavior in the dataset would not produce successful policies. Rather, the learned reward model has to discern the difference between preferred and undesirable behaviors and assign appropriate rewards to trajectories with varying optimality.

## 5.1 MANISKILL TASK SUITE FOR **RQ1**

**Task design.** To test whether our proposed method can mitigate causal confusion in preference-based learning, we design a set of object manipulation tasks in ManiSkill. Each scene has two cubes of different sizes and colors (red vs. blue) on a tabletop, and the agent must manipulate the **larger** cube. We have 4 total tasks: *MS-Pick-Larger*, *MS-Push-Larger*, *MS-Place-Larger*, and *MS-Pull-Larger*. During training, the larger cube is always a fixed color for each task, creating a perfect correlation between color and the correct behavioral choice; The larger cube is always red in *MS-Pick-Larger* and *MS-Pull-Larger*, whereas the larger cube is always blue in *MS-Place-Larger* and *MS-Push-Larger*. At test time we swap the colors so that the distribution shift induces the classic "shortcut" failure: a model that latches onto color will choose the wrong object. Implementation details are described in Appendix A.1.

**Data generation.** We collect synthetic preference queries for each task by pairing trajectories that manipulate the larger and smaller cubes, respectively. These trajectories are generated by training oracle SAC (Haarnoja et al., 2018) policies that manipulate either larger cubes or smaller cubes. Details in the environments and data collection process are introduced in Appendix A.2. Then, the

Table 1: Reward accuracy comparison for ManiSkill (**RQ1**), averaged over 3 seeds. Left block: 2-task setting; right block: 4-task setting. Methods with highest accuracies in each OOD task are bold-faced.

| | 2-task | | | | 4-task | | | | | | | |
| | In Distribution | | Color Swapped | | In Distribution | | | | Color Swapped | | | |
| Model | Pick | Place | Pick | Place | Pick | Push | Place | Pull | Pick | Push | Place | Pull |
| **Single Task** | | | | | | | | | | | | |
| BT | 0.980 | 1.000 | 0.540 | 0.830 | 0.980 | 1.000 | 1.000 | 1.000 | 0.540 | 0.987 | 0.830 | 0.867 |
| **Multi-Task** | | | | | | | | | | | | |
| BT-Multi | 0.953 | 1.000 | 0.600 | 0.820 | 0.987 | 1.000 | 1.000 | 1.000 | 0.707 | **1.000** | 0.840 | 0.907 |
| **Multi-Task with Reasons** | | | | | | | | | | | | |
| RFP | 0.940 | 1.000 | 0.620 | 0.800 | 0.993 | 0.980 | 1.000 | 0.700 | 0.700 | 0.980 | 0.807 | **0.913** |
| ReCouPLe-EC | 0.993 | 1.000 | **0.820** | **0.940** | 1.000 | 1.000 | 1.000 | 1.000 | **0.773** | **1.000** | **0.880** | 0.860 |
| ReCouPLe-IC | 0.967 | 1.000 | 0.633 | 0.807 | 0.993 | 0.987 | 0.993 | 0.993 | 0.600 | **1.000** | 0.807 | 0.867 |

preference label selects the trajectory handling the larger cube. The accompanying rationale $\ell_{reason}$ is *"(because) the cube is larger."* The task label $\ell_{task}$ is simply *"[manipulating verb] the larger cube."* Even though the task text already mentions "larger cube," ReCouPLe uses the rationale as a shared causal axis, projecting trajectories to isolate the size feature in the reason-aligned component while keeping the manipulation verb in a reason-orthogonal, task-specific residual. By contrast, methods without this decomposition often entangle causal (size) and distractor cues (e.g., color), leading to overfitting and failures under color-swap distribution shifts.

**Preference prediction results.** Table 1 shows that performance saturates for all methods on the ID validation split, but clear gaps appear on the *color-swapped* OOD split that flips the color–size correlation. Single-task BT and BT-Multi drop markedly, and a naive reason auxiliary loss (RFP) offers only modest recovery. In contrast, ReCouPLe-EC attains the best OOD accuracy across tasks, with the only exception of *MS-Pull-Larger*. This indicates that ReCouPLe can extract the common causal feature (cube *color*) from tasks with different color–size correlations (e.g., red larger cube in *MS-Pick-Larger*) and blue larger cube in *MS-Place-Larger*)) by projecting trajectory embeddings onto the common reason embedding and isolating such causal features from distractor features (cube *size*). ReCouPLe-IC is comparatively less effective here, since a single causal feature dominates all pairs and the reason-orthogonal components vary little in this suite, and the stricter equality constraint in ReCouPLe-EC yields a clearer advantage.

**Policy Learning Results.** Using the learned rewards to train policies on the OOD environments for 2-task and 4-task settings (Figure 3) mirrors the reward accuracy trends: offline RL datasets with ReCouPLe-trained rewards yield policies with higher success than BT-Multi and RFP. A noticeable advantage is observed in *MS-Pick-Larger* and *MS-Pull-Larger*. Both ReCouPLe-EC and ReCouPLe-IC consistently outperform naive behavior cloning on the play dataset, suggesting that improvements in reward preference prediction translate into downstream policy learning in OOD environments.

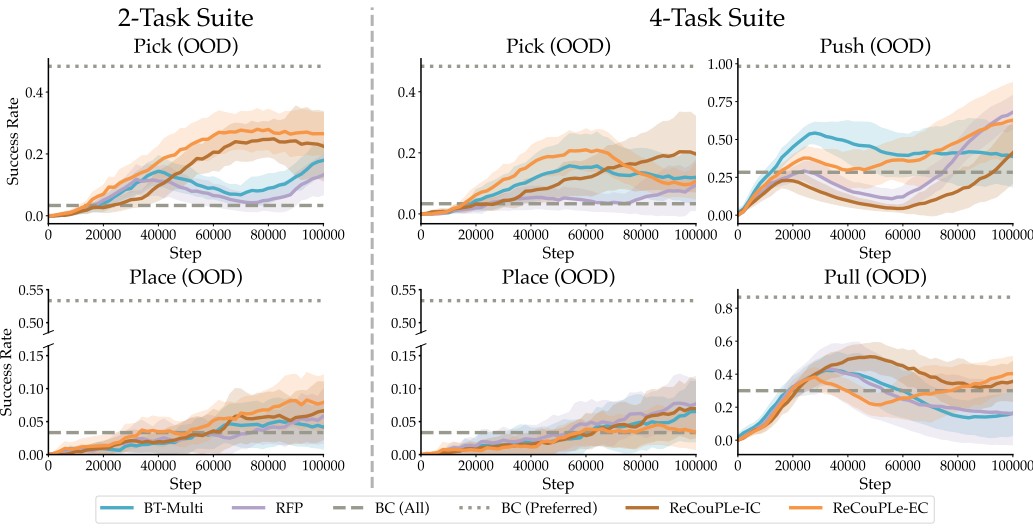

Figure 3: ManiSkill policy learning results, averaged over 3 seeds (mean ± std).

## 5.2 META-WORLD TASK SUITE FOR **RQ2**

**Task design.** We select three training tasks from Meta-World: *Push*, *Push-Wall*, and *Pick-Place-Wall*. We reserve *Pick-Place*, a variant of *Pick-Place-Wall* task without a wall that parallels the structural difference between *Push* and *Push-Wall*. Each task's ground-truth reward is linearly decomposed into interpretable components (grasp, lift, collision avoidance waypoints, etc.) provided by the benchmark.

**Data generation.** We first collect trajectories by rolling out policies with different levels of optimality and Gaussian noise, similar to the data collection procedure in Hejna & Sadigh (2023). Then, for each query, we randomly sample two trajectory segments $\tau_A$ and $\tau_B$ and generate the preference label based on their total reward $\sum_i r(s_i, a_i)$, where Meta-World's pre-defined environment reward can be linearly decomposed into feature components $\{f_j\}$: $r(s,a) = \sum_j w_j f_j(s,a)$. Without loss of generality, suppose $\tau_A$ is preferred over $\tau_B$. Now, we synthetically generate the reason label by computing component-wise advantages $\Delta_j = w_j(f_j(\tau_A) - f_j(\tau_B))$ and convert them to a softmax distribution from which we sample the reason behind the preference:

$$\mathbf{Pr}(\text{choose reason j}) = \frac{\exp(\Delta_j)}{\sum_k \exp(\Delta_k)}$$

Each sampled reason is a free-form sentence such as "keeps a firm grasp while steering toward the goal." Please refer to Appendix B.3 for details. We generate 2000 preference–rationale pairs for each training task (6000 total).

**Preference prediction results.** As demonstrated in Table 2, there is no meaningful difference in validation reward accuracy of training tasks. However, in the novel *Pick-Place* task, both ReCouPLe-IC and ReCouPLe-EC outperform our baselines. This indicates that the reason-aligned subspace learned on the training tasks captures transferable cues for held-out task. ReCouPLe can better transfer preference to novel tasks by decomposing and recomposing semantically grounded features in training tasks (e.g., *Pick-Place-Wall* − (*Push-Wall* − *Push*) ≃ *Pick-Place*).

Table 2: Reward accuracy comparison for Meta-World (**RQ2**), averaged over 3 seeds.

| Model | Training Tasks | | | Novel Task |
|---|---|---|---|---|
| | *Push* | *Push-Wall* | *Pick-Place-Wall* | *Pick-Place* |
| **Multi-Task** | | | | |
| BT-Multi | 0.873 | 0.893 | 0.577 | 0.547 |
| **Multi-Task w/ Reasons** | | | | |
| RFP | 0.870 | **0.900** | 0.647 | 0.553 |
| ReCouPLe-EC | 0.863 | 0.843 | 0.650 | **0.663** |
| ReCouPLe-IC | **0.893** | 0.823 | **0.657** | 0.627 |

**Policy learning results.** We label the rewards of the novel *Pick-Place* task's play data using learned models to assess how well our reward model transfers to the downstream policy learning of a novel task. As indicated in the Figure 4, a similar pattern persists from reward learning results. ReCouPLe can better transfer to a novel task without additional preference queries, compared to both of our baselines. Among our variants for orthogonal consistency loss, ReCouPLe-IC slightly outperforms ReCouPLe-EC in terms of overall preference accuracies. Unlike our ManiSkill experiment, in which preference queries consist of a pair of trajectories with a minimal difference in features other than the stated reasons, datasets in Meta-World contain noisy trajectories with different levels of optimality. Also, each query has a different reason behind its preference. Thus, it is less realistic to assume that reason-orthogonal components should remain identical across compared trajectories. In this setting, the strict equality constraint enforced by ReCouPLe-EC may overly penalize legitimate differences unrelated to the stated reason, harming its performance.

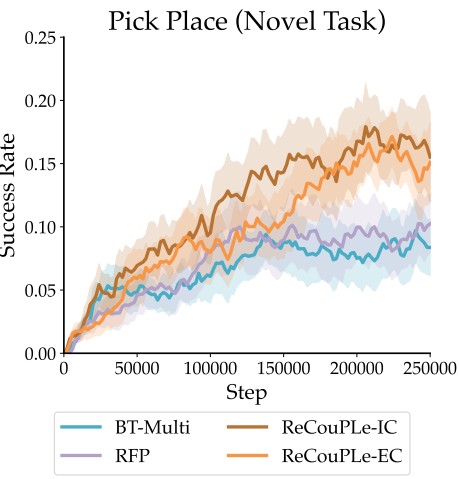

Figure 4: Meta-World policy earning on the held-out task, averaged over 3 seeds (mean ± std). Both ReCouPLe variants outperform the baselines, showing task transfer capability.

## 5.3 ABLATIONS AND ADDITIONAL ANALYSIS

**Ablations.** Table 3 verifies the efficacy of each component in our method: ReCouPLe-no-consistency removes the *consistency* constraint and ReCouPLe-no-consistency-ratio further ablates the *reward-ratio regularizer* in the loss function. Both ablations retain high ID accuracy, but suffer large drops on OOD cases. The consistency loss ensures that preferences are *not* explained by the non-causal, reason-orthogonal component, and the ratio regularizer prevents the trivial collapse

Table 3: Mean reward prediction accuracy over ManiSkill manipulation tasks, averaged over 3 seeds. *ID* and *OOD* refer to in-distribution and out-of-distribution tasks, respectively.

| Model | 2-task | | 4-task | |
|---|---|---|---|---|
| | *ID* | *OOD* | *ID* | *OOD* |
| ReCouPLe | **0.995** | **0.872** | **1.000** | **0.878** |
| ReCouPLe-no-consistency | 0.980 | 0.726 | 0.977 | 0.745 |
| ReCouPLe-no-consistency-no-ratio | 0.987 | 0.727 | 0.990 | 0.730 |

of this component. Together, they account for ReCouPLe's robustness gains under distribution shift.

**Effectiveness of ReCouPLe in image-based control tasks.** While our main experiments focus on state-based control, previous works demonstrate that manipulation tasks are especially vulnerable to causal confusion from distribution shifts in non-causal visual features (Park et al., 2021). To address this, we evaluate ReCouPLe on image-based tasks across both ManiSkill and a challenging Meta-World visual suite. Table 4 shows the ManiSkill results: with raw visual inputs, all baselines yield significantly lower reward prediction accuracy on the out-of-distribution (OOD) validation set, as their reward models overly attend to spurious features (cube *color*). RFP raises OOD accuracy only slightly, confirming that naively adding an auxiliary BCE loss for reasons provides an informative but insufficient signal. In contrast, ReCouPLe significantly improves OOD accuracy and outperforms all baselines. Under the 4-task setting, both ReCouPLe variants reach near-perfect accuracies ($\geq 0.96$).

Table 4: Reward accuracy comparison for in image-based ManiSkill (**RQ1**) environment, averaged over 3 seeds. Methods with highest accuracies in each OOD task are bold-faced.

| | 2-task | | | | 4-task | | | | | | | |
|---|---|---|---|---|---|---|---|---|---|---|---|---|
| | In Distribution | | Color Swapped | | In Distribution | | | | Color Swapped | | | |
| Model | Pick | Push | Pick | Push | Pick | Push | Place | Pull | Pick | Push | Place | Pull |
| **Single Task** | | | | | | | | | | | | |
| BT | 0.833 | 1.000 | 0.167 | 0.610 | 0.833 | 1.000 | 0.980 | 1.000 | 0.167 | 0.610 | 0.460 | 0.053 |
| **Multi-Task** | | | | | | | | | | | | |
| BT-Multi | 0.870 | 0.999 | 0.167 | 0.673 | 0.867 | 1.000 | 0.987 | 1.000 | 0.533 | 0.867 | 0.833 | 0.587 |
| **Multi-Task with Reasons** | | | | | | | | | | | | |
| RFP | 0.847 | 0.990 | 0.290 | 0.813 | 0.867 | 1.000 | 0.993 | 1.000 | 0.807 | 0.967 | 0.947 | 0.833 |
| ReCouPLe-EC ($\lambda_{ratio} = 0.4$) | 0.980 | 1.000 | **0.707** | **0.987** | 0.980 | 1.000 | 0.993 | 1.000 | **0.960** | **1.000** | **1.000** | 0.980 |
| ReCouPLe-IC ($\lambda_{ratio} = 0.4$) | 0.940 | 1.000 | 0.433 | 0.927 | 0.960 | 1.000 | 1.000 | 1.000 | 0.960 | **1.000** | 0.993 | **0.987** |

To test robustness in more challenging settings, we designed a Meta-World experiment (*Push* and *Push-Wall*) introducing background color as a non-causal distractor. For *Push*, preferred trajectories were collected in environments with a blue background, while non-preferred trajectories with comparably more suboptimal behavior had a gray background. For *Push-Wall*, this color scheme was flipped (see Figure 6 in Appendix C for a visual overview). We evaluated the reward models based on their accuracy in correctly predicting preferences between trajectory pairs in an OOD environment where these background colors were swapped. We maintained diverse rationales across tasks (4 reasons for *Push*, 5 for *Push-Wall*) sharing high-level semantics, sampled from a softmax distribution of component-wise advantages identical to the generation process in Section 5.2. As shown in Table 5, the reward prediction accuracy of standard baselines collapse when visual confounders are swapped (e.g., BT baseline drops to 0.233 on *Push* OOD). However, ReCouPLe-IC successfully disentangles the shared causal reasoning from the spurious background color and demonstrates robustness under the distribution shift, maintaining accuracies of 0.793 on *Push* and 0.900 on *Push-Wall*.

Table 5: Reward accuracy comparison for custom visual causal confusion tasks in Meta-World. ReCouPLe successfully avoids overfitting to spurious background colors under diverse reasoning.

| Model | Push (Visual OOD) | Push-Wall (Visual OOD) |
|---|---|---|
| BT (Baseline) | 0.233 | 0.113 |
| BT-Multi | 0.367 | 0.227 |
| ReCouPLe-EC | 0.713 | 0.760 |
| ReCouPLe-IC | **0.793** | **0.900** |

**Robustness to Linguistic Diversity.** A common concern with language-conditioned reward models is their tendency to overfit to specific, simplistic instruction templates. To verify that ReCouPLe learns robust semantic representations rather than overfitting to specific phrasing, we evaluated its robustness to phrasing diversity in reason labels. We replaced the single canonical reason ("*cube is larger*") with 16 distinct paraphrases per task, incorporating synonyms ("*cube is bigger*"), specific descriptors ("*red cube is larger*"), passive voice ("*larger cube is picked*"), and negations ("*smaller cube is not picked*"). Notably, training with diversified rationales actually improved OOD generalization: ReCouPLe-EC's reward accuracies on the *Pick* (OOD) task increased from 0.820 with original to 0.867 with diversified rationales. This result suggests that our method is not brittle to specific sentence structures. It demonstrates the trajectory encoder successfully extracts common causal features and aligns them with lexically different but semantically similar reasons, rather than memorizing exact text strings. We detail the reason labels and full results in Appendix D.

**Scalability to Sparse Explanations.** Requiring a natural language rationale for every preference annotation can be prohibitively expensive in large-scale offline datasets. To evaluate ReCouPLe's label efficiency, we trained the reward model on datasets where rationales were provided for only a subset (25% and 50%) of the preference pairs. For the remaining pairs, the model relied solely on the standard binary cross-entropy loss without rationale alignment. Remarkably, ReCouPLe proved highly label-efficient. Even when 75% of reasons were missing (the 25% subset), ReCouPLe-EC retained a robust out-of-distribution reward accuracy of 0.783 on the *Pick* OOD task, vastly outperforming the standard BT baseline (0.540) which lacked reason labels entirely. This indicates that the causal signal derived from a small fraction of rationales successfully propagates through the shared trajectory encoder to regularize the entire dataset. Full results for this sparse setting are detailed in Appendix E.

**Impact of Preference Sample Size.** Beyond the sparsity of rationales, we also analyzed how the absolute volume of preference queries impacts downstream policy robustness. We conducted an extensive scaling experiment on the 2-task ManiSkill setup, varying the dataset size from 200 to 2,000 queries per task. Note that the original experiment in Section 5.1 had 1000 preference queries per task. We observed that baseline methods, such as BT-Multi and RFP, show only marginal improvements as data increases and quickly hit a performance ceiling. In contrast, ReCouPLe-EC successfully leverages the additional preference queries. By anchoring the reward function to the provided rationale, the model uses the extra samples to sharpen the distinction between the causal feature and the confounders. Consequently, the performance steadily improves with dataset size, with its reward accuracy on the *Pick* (OOD) task improving to 0.913 at 2,000 queries. The full scaling analysis and tables are provided in Appendix F.

## 6 CONCLUSION

**Summary.** We introduce ReCouPLe, a lightweight preference learning method that is robust against causal confusion and effective for task transfer by leveraging natural language reasons behind preferences. ReCouPLe turns free-form language rationales into causal projection axes for preference-based reward learning, and uses them to separate out the part of the trajectory that explains the preference, ensuring the model focuses on the feature that actually matters. Furthermore, ReCouPLe leverages shared causal structure across multiple tasks to transfer reward signals without additional preference data or language model fine-tuning. Across two complementary evaluations, ReCouPLe consistently mitigates causal confusion and exhibits strong zero-shot transfer to novel tasks.

**Limitations and Future Work.** While ReCouPLe shows strong gains, some limitations remain. Our method assumes that rationales are easily available and reliable, yet in practice they may be noisy or costly to obtain. A promising future direction is to explore active learning strategies that selectively query rationales when they are expected to provide the most causal signal. Another limitation is that our evaluation is restricted to simulated domains; validating ReCouPLe in real-world robotic settings will be essential to assess its robustness under natural human input. Finally, our current framework focuses on reward modeling, leaving an additional RL step for policy learning. Extending it to directly learn policies from rationales and preferences (An et al., 2023; Rafailov et al., 2023; Hejna et al., 2024), bypassing the reward-learning loop, could further improve efficiency and practicality.

## REPRODUCIBILITY STATEMENT

To ensure reproducibility, we provide the source code in the supplementary material and publicly on GitHub at `https://github.com/mj-hwang/ReCouPLe`. We include a README file containing commands for all experiments. We describe the evaluation environment and training details in the Appendix.

## LLM USAGE

We only used large language models (LLMs) to assist with grammar correction and rewording. No model-generated content was used for scientific claims, experiments, or core contributions. All ideas and analyses are original and solely developed by the authors.

## ACKNOWLEDGEMENTS

We thank the USC Center for Advanced Research Computing (CARC) for providing us with compute resources. YK and EB acknowledge funding by the Airbus Institute for Engineering Research (AIER) for this project.

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

# APPENDIX

## Table of Contents

## A  MANISKILL EXPERIMENTAL DETAILS

### A.1  DETAILS AND IMPLEMENTATION OF TASKS

As introduced in Section 5.1, we design 4 custom manipulation environments (*MS-Pick-Larger*, *MS-Push-Larger*, *MS-Place-Larger*, and *MS-Pull-Larger*) in which each scene has two cubes of different sizes and colors by modifying a set of tabletop manipulation tasks provided in ManiSkill (Tao et al., 2025). All the original tasks require the robot to manipulate a single cube or sphere, but we instead put two cubes with different sizes and colors in the scene to entangle a causal feature (object size) with a non-causal distractor feature (object color), as shown in Table 6. The top row of Figure 5 displays exemplar terminal states for each task, where the robot successfully manipulates the larger cube.

Table 6: Task descriptions and settings for ManiSkill tasks.

| Task | Task Description | Larger Cube Color | Smaller Cube Color |
|------|-----------------|:-----------------:|:------------------:|
| *MS-Pick-Larger* | "pick up larger cube to target sphere" | Red | Blue |
| *MS-Place-Larger* | "place larger cube in target bin" | Blue | Red |
| *MS-Push-Larger* | "push larger cube toward target line" | Blue | Red |
| *MS-Pull-Larger* | "pull larger cube toward green line" | Red | Blue |

For all tasks, we randomize the initial pose of two cubes and the target object. Additionally, for each task, we design two versions of ground-truth reward function that incentivizes the robot to manipulate either the larger cube or the smaller cube. This is later used for training expert RL policies with which we collect trajectories in preference dataset.

**Observation modalities.** For state-based experiments in Section 5.1, we design a compact state that includes, for each of the two cubes, its *color*, *size*, and *pose*. To avoid trivial dependence on input ordering, we *randomize the order of the two cubes* in the concatenated state at every episode. This forces methods to identify the correct causal feature (size) rather than relying on position/order or color; it thus mirrors the causal-confusion risk present in visuomotor policy learning. We then concatenate this with the robot proprioception and the target object pose to create the state. For image-based experiments in Section 5.3, we use a raw 128×128 RGB frame as an input modality.

| *MS-Pick-Larger* | *MS-Place-Larger* | *MS-Push-Larger* | *MS-Pull-Larger* |
|---|---|---|---|

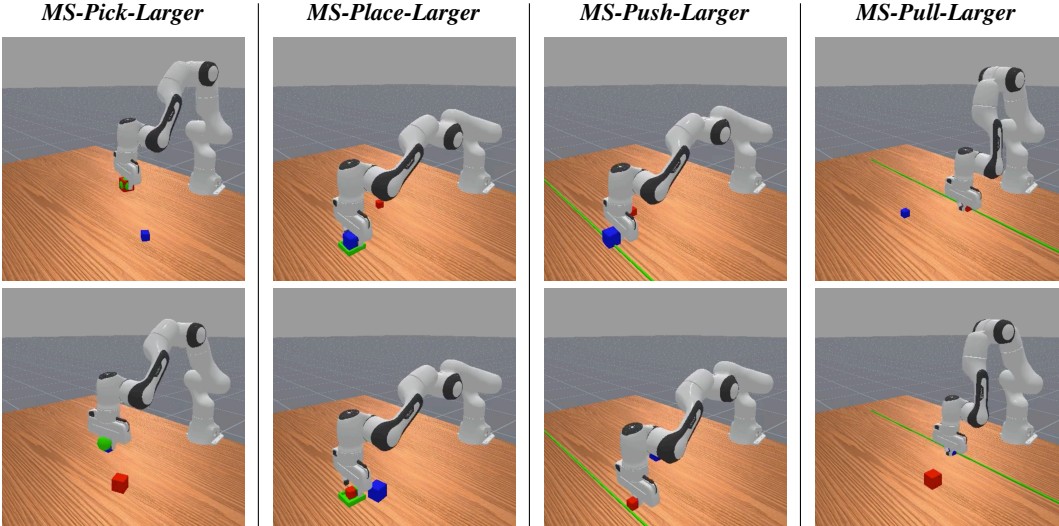

Figure 5: Terminal states for custom ManiSkill tasks. Each column represents a specific task, with the **top row** showing preferred trajectories (manipulating the larger cube) and the **bottom row** showing non-preferred trajectories (manipulating the smaller cube). Tasks and their respective color confounder are defined in Table 6.

## A.2 DATA COLLECTION DETAILS

For the state-based experiments (Section 5.1), we train SAC (Haarnoja et al., 2018) policies that either manipulate the larger cube or the smaller cube, for each manipulation task. The former policy is used to collect preferred trajectories, and the latter is used to collect trajectories that are not preferred. Then, for each task, we generate 1000 synthetic preference pairs by pairing trajectory segments that manipulate the larger and smaller cubes, respectively. These segments are sub-sampled from the oracle's collected trajectories, with the length of 4. For the image-based experiments (Section 5.3), we design motion planning solutions that manipulate either the larger or the smaller cube for each task due to higher sample complexity in training visuomotor RL policies. Similar to the state-based experiments, we generate 500 synthetic preference pairs for each task, where the sub-sampled segment length is 64. The terminal states of the full trajectories from which these segments are sub-sampled are visualized in Figure 5.

## B META-WORLD EXPERIMENTAL DETAILS

### B.1 DATA COLLECTION

In this section, we provide details on how offline datasets for Meta-World tasks are constructed. Similar to Hejna & Sadigh (2023), we first collect trajectories with varying levels of optimality and behavior for each task. These ground-truth policies are scripted, inverse-kinematics-based expert policies provided by the Meta-World benchmark (Yu et al., 2020). The collection scheme is as follows:

- 400 trajectories collected with the ground-truth policy for the target task, with varying levels of random noise. We use Gaussian noise with standard deviations [0.1, 0.3, 0.5].
- 100 random trajectories with *uniform random actions*.
- 100 suboptimal trajectories collected with the ground-truth policy, but for a *different random initialization* in each target environment.
- 100 suboptimal trajectories collected with the ground-truth policy for a *different task*, other than each target environment. For tasks without a wall in the center (i.e., *Pick-Place* and *Push*),

These data are generated using the scripted, inverse-kinematics-based expert policies provided by the Meta-World benchmark. We over-sample trajectories with the noisy ground-truth policy, in order to

guarantee diversity in sampled reasons. If we only collect substantially suboptimal or nearly random trajectories, we empirically observed that it significantly limits the diversity of reasons; reasons that are related to near-goal states are barely sampled as most trajectories achieve limited task progression.

After we collect trajectories, we construct our preference dataset with the following procedure. We first uniformly sample segments with sub-trajectory length 32 from the collected trajectories. For each task, we sample 4000 segments and create 2000 preference pairs. Assignment of preference labels and corresponding reason texts is described in Section 5.2. This process results in 6000 total preference pairs, as we have three training tasks (*Pick-Place-Wall*, *Push*, *Push-Wall*).

For our held-out validation dataset for reward accuracy evaluation, we simply repeat the same process as training dataset. We however collect 100 preference pairs per task, which results in 400 total preference pairs (3 training tasks and 1 novel task). We also collect play data (i.e., offline RL data without reward label) in the same process. For each task, we collect 2000 preference pairs, which we later label reward values with our learned reward models and train offline RL policies with.

## B.2 META-WORLD NATURAL LANGUAGE TASK DESCRIPTIONS

Table 7 has a compiled list of task descriptions $\ell_{\text{task}}$. As described in Section 3.2, we convert these task descriptions to create task embeddings $\theta$.

Table 7: Natural language task descriptions for Meta-World tasks.

| Task | Task Description |
|------|-----------------|
| *Push* | "make contact and push puck to goal" |
| *Push-Wall* | "make contact, bypass wall via waypoint, and push puck to goal" |
| *Pick-Place* | "pick up puck, lift it, and place it on goal" |
| *Pick-Place-Wall* | "pick up puck, use waypoint to bypass wall, and place it on goal" |

## B.3 META-WORLD NATURAL LANGUAGE RATIONALES

Table 8 has a compiled list of reasons $\ell_{\text{reason}}$. As described in the data generation section of Section 5.2, we sample the reason behind the preference from softmax distribution of component-wise advantages. Here, each reason corresponds to an appropriate feature component in the total reward function, where the environment reward can be linearly decomposed into interpretable feature components. This decomposition is based on the reward formulation of Meta-world environments Yu et al. (2020).

Table 8: Reason codes used per task.

| Code | Reason Text |
|------|-------------|
| *Push* | "pushes more decisively after making contact" |
| *Push* | "pushes puck closer to goal" |
| *Push-Wall* | "pushes puck toward waypoint" |
| *Push-Wall* | "guides puck past wall" |
| *Push-Wall* | "pushes puck toward goal after clearing wall" |
| *Push & Push-Wall* | "maintains firm grip on puck" |
| *Push & Push-Wall* | "makes contact with puck sooner" |
| *Pick-Place* | "keeps firm grip while moving puck toward goal" |
| *Pick-Place* | "carries puck toward goal while lifted" |
| *Pick-Place-Wall* | "keeps firm grip while moving puck toward waypoint" |
| *Pick-Place-Wall* | "carries puck toward waypoint while lifted" |
| *Pick-Place-Wall* | "clears wall with puck lifted" |
| *Pick-Place-Wall* | "carries puck toward goal after clearing wall" |
| *Pick-Place & Pick-Place-Wall* | "lifts puck cleanly" |
| All tasks | "finishes at goal spot" |

### B.4 JUSTIFICATION OF RATIONALE EXTRACTION AND SOFTMAX SAMPLING

As described in Section 5.2, we sample the reason behind the preference from a softmax distribution of component-wise advantages, rather than from a deterministic argmax. This design choice is grounded in Discrete Choice Theory, specifically the Random Utility Model (McFadden, 1972) and the Luce Choice Axiom (Luce, 1959). In a human-in-the-loop setting, users are probabilistic; while they likely cite the dominant factor, they occasionally cite secondary relevant factors. Sampling from the softmax simulates a realistic user providing reasons proportional to their saliency. Using an argmax would assume a "perfect" annotator, artificially reducing the diversity of the training signal.

## C DETAILS ON META-WORLD CUSTOM VISUAL CAUSAL CONFUSION TASKS

We provide additional implementation details for the custom Meta-World environments introduced in Section 5.3. Similar to the data collection scheme in Appendix B, we use the scripted expert policies (Yu et al., 2020) with different levels of Gaussian noise. Specifically, preferred trajectories were collected with Gaussian noise of $\epsilon = 0.1$, while non-preferred trajectories were collected with Gaussian noise of $\epsilon = 1.0$, resulting in significantly more suboptimal behavior. By design, the background floor color serves as a non-causal distractor that is perfectly correlated with these preference labels during training (*e.g.*, blue for preferred and gray for non-preferred in *Push*, as shown in Figure 6). This setup allows us to evaluate whether reward models rely on these spurious visual cues or instead capture the underlying task completion.

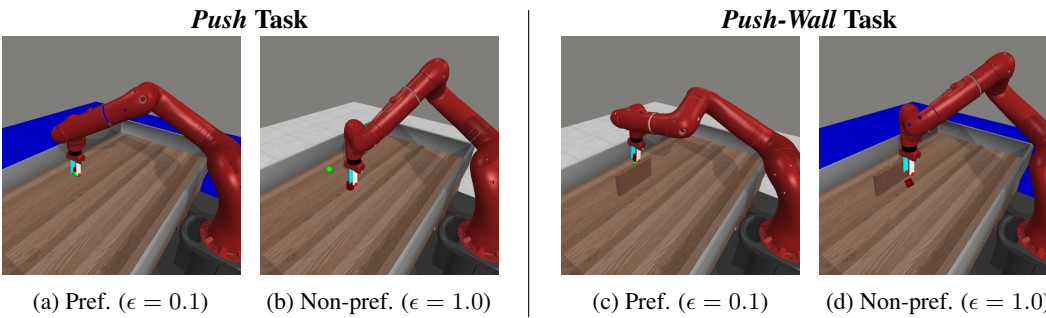

| *Push* Task | | *Push-Wall* Task | |
|---|---|---|---|
| (a) Pref. ($\epsilon = 0.1$) | (b) Non-pref. ($\epsilon = 1.0$) | (c) Pref. ($\epsilon = 0.1$) | (d) Non-pref. ($\epsilon = 1.0$) |

Figure 6: Visualizations of terminal states from the custom visual causal confusion environments. During training, background color is perfectly correlated with trajectory preference. Robustness is evaluated on an OOD distribution where this mapping is inverted, requiring the reward model to generalize beyond spurious visual features.

## D ROBUSTNESS TO LINGUISTIC DIVERSITY

To verify that our approach generalizes across different natural language expressions rather than overfitting to specific rationale phrases, we assess ReCouPLe's robustness to phrasing and semantic diversity. In the original ManiSkill experiments, a single canonical reason was used (Section 5.1). We generated 16 distinct paraphrases per task to replace the single canonical reason. These variations included:

- **Synonyms:** "cube is bigger", "object is larger"
- **Specific Descriptors:** "red cube is larger", "blue object is bigger"
- **Passive Voice:** "larger cube is picked"
- **Negation/Contrast:** "smaller cube is not picked", "tinier object is not placed"

As shown in Table 9, training with diversified rationales actually improves OOD performance. This demonstrates that the trajectory encoder successfully extracts common causal features and aligns them with semantic representations shared across varied phrasing, rather than relying on simple pattern matching.

Table 9: Robustness to Rationale Diversity (ManiSkill OOD Reward Accuracies)

| Model | Pick (OOD) | Place (OOD) |
|---|---|---|
| BT (Baseline) | 0.540 | 0.830 |
| ReCouPLe-EC (Original) | 0.820 | 0.940 |
| ReCouPLe-EC (Diversified) | **0.867** | **0.967** |
| ReCouPLe-IC (Original) | 0.633 | 0.807 |
| ReCouPLe-IC (Diversified) | 0.620 | 0.820 |

# E  SCALABILITY TO SPARSE EXPLANATIONS

Annotating every preference pair with a rationale can be expensive. We investigated our method's label efficiency by training on datasets where rationales were provided for only a subset (25% and 50%) of the preference pairs. For pairs without reasons, we simply minimized the standard binary cross-entropy loss following the Bradley-Terry model. As shown in Table 10, ReCouPLe is highly label-efficient: even when 75% of reasons are missing, the causal signal from a small set of rationales propagates to the entire dataset.

Table 10: Scalability to Sparse Explanations (ManiSkill OOD Reward Accuracies)

| Model | Pick (OOD) | Place (OOD) |
|---|---|---|
| BT (Baseline) | 0.540 | 0.830 |
| ReCouPLe-EC (25% Reasons) | **0.783** | 0.940 |
| ReCouPLe-EC (50% Reasons) | 0.767 | **0.947** |
| ReCouPLe-IC (25% Reasons) | 0.673 | 0.807 |
| ReCouPLe-IC (50% Reasons) | 0.660 | 0.847 |

# F  IMPACT OF PREFERENCE SAMPLE SIZE

To address how preference sample size impacts our method, we conducted an extensive scaling experiment on the 2-task ManiSkill setup, varying the dataset size from 200 to 2,000 queries per task. While baseline methods like BT-Multi and RFP show marginal improvements as data increases, they quickly hit a performance ceiling because adding more samples merely reinforces their reliance on spurious features. In contrast, ReCouPLe-EC successfully leverages additional data to sharpen the distinction between causal features and confounders.

Table 11: Scaling Analysis on ManiSkill (2-Task Setup). The 1000-query column corresponds to the original dataset size used in our main experiments in Section 5.1

| Preference Queries | 200 | 500 | 1000 | 2000 |
|---|---|---|---|---|
| **Task: Pick (OOD)** | | | | |
| BT-Multi | 0.573 | 0.627 | 0.600 | 0.693 |
| RFP | 0.540 | 0.613 | 0.620 | 0.760 |
| ReCouPLe-EC | **0.733** | **0.820** | **0.820** | **0.913** |
| ReCouPLe-IC | 0.547 | 0.580 | 0.633 | 0.793 |
| **Task: Place (OOD)** | | | | |
| BT-Multi | 0.667 | 0.760 | 0.820 | 0.767 |
| RFP | 0.733 | 0.720 | 0.800 | 0.760 |
| ReCouPLe-EC | 0.747 | **0.847** | **0.940** | **0.920** |
| ReCouPLe-IC | **0.773** | 0.793 | 0.807 | 0.807 |

## G    CORRELATION BETWEEN REWARD ACCURACY AND POLICY PERFORMANCE

In standard reward learning, high reward model accuracy does not always translate to better policies due to "Causal Goodhart" effects (Manheim & Garrabrant, 2018; Gao et al., 2023), where a proxy reward model relies on features that correlate with the gold label but do not cause it. ReCouPLe is explicitly designed to address this by using the rationale to project the reward onto the causal axis, forcing the model to ignore spurious features. Consequently, OOD reward accuracy becomes a reliable binary detector of causal disentanglement. Our empirical results confirm this: we observed a Pearson correlation of 0.924 and a Mean Spearman's Rank correlation of 0.933 between OOD reward accuracy and policy success in Meta-World tasks in Section 5.2.

