# OpenReview forum: "Causally Robust Reward Learning from Reason-Augmented Preference Feedback"
_ICLR.cc/2026/Conference — ICLR 2026 Poster_

### Official Review · Reviewer_NmL7 · 2025-10-26

**Soundness:** 2
**Presentation:** 3
**Contribution:** 3
**Rating:** 6
**Confidence:** 3

**Summary:**

This paper considers the problem of reward learning in PbRL, where the data can be confounded. The authors propose ReCouPLe, which use natural language task description/explanations to resolve such spurious correlations. Specifically, the rewards are decomposed into two components (i.e., a component parallel to the reason and an orthogonal component), and only the parallel component is used to capture preferences via the BT model. The orthogonal component (non-causal feature) is then penalized to be consistent between trajectories. Experiments show that the proposed approach outperforms standard baselines that do not utilize the auxiliary information.

**Strengths:**

The paper is clearly written and easy to follow. The environments and datasets are carefully designed to capture confounding problems in PbRL and the experiments clearly demonstrate the effectiveness of the proposed approach. Although it is not immediately clear how frequently causal confusions occur in real-world settings, I believe the method has potential applications beyond the specific cases studied here. In particular, it could be useful for more general problems where explanations are available during reward modeling (e.g., Table 4 shows that in-distribution tasks can also benefit from the proposed method), making it of interest to the broader community.

**Weaknesses:**

My main concern is that it is well-known that language models can be sensitive to phrasing, and this is not really explored in the paper (e.g., using different LMs, paraphrasing).
It is also not clear what kind of LMs should be used to best extract the task/reason embeddings.
For Fig 3, I think it would be nice to have in-distribution performances also reported; otherwise, it is difficult to tell if the performances stagnate because of problems from RL training or reward modeling.
Aside from the ablations presented in Sec 5.3, I would suggest the authors also ablate on the LM side to demonstrate the robustness of the method.

**Questions:**

1. Line 393: This sampling distribution may lead to completely incorrect reasons given trajectory pairs. Can the authors elaborate on this design choice?
2. Line 409: What does it mean to decompose and recompose semantically? Can the authors provide additional evidence for this behavior?
3. How sensitive is the method regarding task descriptions / reason codes?

---

> ### Author Response · Authors · 2025-12-03
> **Response to Reviewer NmL7 (1/2)**
>
> ### **\[W, Q-3\] Robustness to Phrasing and Task Descriptions**
>
> A primary concern raised by the reviewer is the sensitivity of Language Models to phrasing. To address this, we conducted additional experiments on the ManiSkill Pick/Place tasks during the rebuttal phase.
>
> **Experimental Setup:** In the original setup, a single canonical reason was used. To verify linguistic robustness to phrasing and semantic diversity, we generated **16 distinct paraphrases** per task. These variations included:
>
> * **Synonyms:** *"cube is bigger"*, *"object is larger"*
> * **Specific Descriptors:** *"red cube is larger"*, *"blue object is bigger"*
> * **Passive Voice:** *"larger cube is picked"*
> * **Negation/Contrast:** *"smaller cube is not picked"*, *"tinier object is not placed"*
>
> We trained ReCouPLe-EC using these randomized paraphrases (ReCouPLe-EC Diversified).
>
> **Results:** As shown in Table 1, using diverse paraphrases resulted in reward accuracies comparable to, and in some OOD cases slightly better than, the fixed-reason baseline (e.g., 0.820 $\rightarrow$ 0.867 for **ReCouPLe-EC** in Pick tasks). These results suggest that our method is not brittle to specific sentence structures. We hypothesize that training on diverse paraphrases prevents the model from overfitting to the specific embedding artifacts of a single static sentence. Instead, the variance in the text embeddings likely serves as a regularizer, encouraging the model to learn a direction in the latent space that is robust to linguistic perturbations.
>
> *Table 1: Robustness to Rationale Diversity (Reward Accuracy on on ManiSkill OOD)*
>
> | Model | Pick (OOD) | Place (OOD) |
> | :---- | :---- | :---- |
> | BT (Baseline) | 0.540 | 0.830 |
> | **ReCouPLe-EC (Original)** | 0.820 | 0.940 |
> | **ReCouPLe-EC (Diversified)** | **0.867** | **0.967** |
> | **ReCouPLe-IC (Original)** | 0.633 | 0.807 |
> | **ReCouPLe-IC (Diversified)** | 0.620 | 0.820 |
>
> ### **\[W\] Scalability and Robustness to Sparse Explanations**
>
> The reviewer noted that our method is particularly useful when explanations are available. To further demonstrate the scalability of our approach and quantify its dependence on dense feedback, we investigated robustness when reason feedback is provided for only a subset of preference pairs (25% and 50%).
>
> **Results:** Even when rationales are available for only **25%** of the training pairs, ReCouPLe-EC maintains significantly more robust OOD accuracy (0.783) compared to the standard BT baseline (0.540). This indicates that ReCouPLe does not strictly require a rationale for every data point; the causal constraints applied to the rationale-annotated subset appear sufficient to shape the trajectory representation and prevent collapse into spurious correlations. This suggests the method remains effective even in settings where acquiring full language feedback is impractical.
>
> *Table 2: Scalability to Sparse Explanations (Reward Accuracy on ManiSkill OOD)*
>
> | Model | Pick (OOD) | Place (OOD) |
> | :---- | :---- | :---- |
> | BT (Baseline) | 0.540 | 0.830 |
> | **ReCouPLe-EC (25% Reasons)** | **0.783** | **0.940** |
> | **ReCouPLe-EC (50% Reasons)** | 0.767 | 0.947 |
> | **ReCouPLe-IC (25% Reasons)** | 0.673 | 0.807 |
> | **ReCouPLe-IC (50% Reasons)** | 0.660 | 0.847 |

---

> > ### Author Response · Authors · 2025-12-03
> > **Response to Reviewer NmL7 (2/2)**
> >
> > ### **\[W\] Rationale for Choosing T5 over BERT**
> >
> > Regarding why we selected T5 over other architectures like BERT, our decision was driven by two main factors:
> >
> > 1. **Architectural Suitability:** BERT \[1\] is optimized for token-level masked language modeling, often requiring specific pooling strategies to yield coherent sentence embeddings. In contrast, T5 \[2\] is trained on a sequence-to-sequence objective naturally aligned with processing complete statements. This architecture yields better semantic clustering for full sentences, which is crucial for our method where the "reason" is a complete sentence describing a behavior.
> > 2. **Empirical Performance in Reward Learning:** There are several previous works \[3, 4\] that aim to learn a trajectory encoder aligned with comparative language feedback for trajectory improvement. Their analyses demonstrate that pretrained T5 embeddings can encode necessary semantic information to which a trajectory encoder captures and aligns relevant features present in trajectories.
> >
> > ### **\[Q-1\] Clarification on Sampling Distribution (Line 393\)**
> >
> > The reviewer asked about the design choice to sample reasons from a softmax distribution. This is grounded in Discrete Choice Theory \[5, 6\]. In a human-in-the-loop setting, users are probabilistic; while they likely cite the dominant factor, they occasionally cite secondary relevant factors. Sampling from the softmax simulates a realistic user providing reasons proportional to their saliency. Using argmax would assume a "perfect" annotator, artificially reducing the diversity of the training signal.
> >
> > ### **\[Q-2\] Decomposition and Recomposition (Line 409\)**
> >
> > Finally, regarding the semantic meaning of "decompose and recompose," we clarify that these describe geometric operations in the reward space:
> >
> > * **Decompose:** Projecting the trajectory representation $\phi(s)$ onto the language embedding axis to mathematically separates the **causal component** ($\phi_{\parallel}$; parallel to reason) from the **confounding component** ($\phi_{\perp}$; orthogonal). This allows the reward model to isolate
> > the causal feature specified in the rationale to explain the user’s preference with the Bradley-Terry model, while preventing non-causal features to affect the pairwise preference as much.
> > * **Recompose:** Summing these components ($r_{\parallel} + r_{\perp}$) to form the final reward signal for the agent. We explicitly retain the orthogonal component as it can capture essential task structures (e.g., goal-reaching or safety constraints) that are common to both trajectories. While these features might not drive the specific pairwise preference since they do not differ between the pair, they remain critical for the agent to learn a successful policy.
> >
> > \[1\] Devlin et al., BERT: Pre-training of Deep Bidirectional Transformers for Language Understanding, NAACL 2019
> > \[2\] Raffel et al., Exploring the Limits of Transfer Learning with a Unified Text-to-Text Transformer, JMLR 2020
> > \[3\] Yang et al., Learning from Comparative Language Feedback, CoRL 2024
> > \[4\] Hirota et al., Active Reward Learning and Iterative Trajectory Improvement from Comparative Language Feedback, IJRR 2025
> > \[5\] McFadden, Conditional Logit Analysis of Qualitative Choice Behavior, 1973
> > \[6\] Luce, Individual Choice Behavior: A Theoretical Analysis, 1959

---

### Official Review · Reviewer_fFvS · 2025-10-27

**Soundness:** 2
**Presentation:** 3
**Contribution:** 3
**Rating:** 2
**Confidence:** 4

**Summary:**

The paper presents a method to incorporate preference rationales into the learning of the trajectory representation used by the reward model to assign rewards per state-action pair. The method relies on the presence of a language-based task description that is embedded and then compared to each state-action pair embedding from the given trajectory. The trajectory learning approach combines the current representation of each state-action pair in a trajectory with an embedding of the provided preference rationale, and relies on a representation of the trajectory that excludes those features most relevant to the preference rationale. The training objective then combines three objectives that aim to: (1) force the state-action pair embedding to focus on rationale-relevant features, (2) ignore features not relevant to the rationale, and (3) prevent a collapse of the trajectory representation. The experiments are run on two different domains, ManiSkill and Metaworld, each of which have different levels of difficulty and rationale complexity. The results suggest the proposed method, ReCouPLE, out performs baselines on reward modelling and learned-policy quality.

**Strengths:**

- The paper applies a method that has been used in goal-conditioned RL to the preference learning setting to boost the limited amount of information available in a preference label.
- Addressing the limited amount of information available in a preference signal is a key problem to solve improve general field ability to learn effective and robust reward models.
- The paper is easy to read and follow.

**Weaknesses:**

- Some experiments are missing to truly understand where ReCouPLE performs well:
     - impact of number of preference samples on reward and learned-policy quality
     - impact of noisy preference labels
     - impact of noisy rationales
     - combining data with and without rationales as, in practice, collecting large datasets with rationales will be expensive and impractical
- The ManiSkill experiments lack diversity in rationales, so it is not clear how well the results will generalize.
- While one of ReCouPLe-EC or -IC typically are a best performer, frequently one or more baseline is better than at least one of the two versions, and this varies by task for ManiSkill. This suggests the method is fragile and task-specific. There is no discussion of what might lead to this performance sensitivity. Given the sensitivity of ReCouPLEe performance to the task in ManiSkill, there should be more test tasks for MetaWorld.
- There should be MetaWorld test tasks less closely related to the training tasks to understand how well the method works in more challenging and realistic multi-task settings.
- It is not possible to assess how impact or meaningful the ReCouPLe performance gains are. When presenting the test results for policy training, results should be provided when training on the true reward. For example, if the training on the true reward provides a success rate of 0.8, a success rate difference of 0.07 is less meaningful.
- For the ablations and analysis results in Section 5.3 (ablations and image-based control tasks), results should be provided for policy learning, not only reward model accuracy, especially as the reward model accuracy score differences are large. While there may be a difference in reward model accuracy, it is important to understand how much of an impact that difference has on learned-policy performance.

**Questions:**

- How much data is used in the ManiSkill experiments?
- What is the strength of correlation between the reward model's performance and policy performance? This important to answer as there is evidence on the LLM side of reward learning and PbRL that reward model preference-labelling performance is not always strongly predictive of how well the reward can be used to train a policy. This makes reward model selection challenging.
- Why is the color swap reward model accuracy higher in Table 1 for 4-task + pull + RFP?
- How do the baseline methods compare ReCouPLe in terms of number of learnable parameters? This would speak to the generally upper bound on method expressivity, which impacts general overall performance.

---

> ### Author Response · Authors · 2025-12-03
> **Response to Reviewer fFvS (1/3)**
>
> We thank the reviewer for the thoughtful assessment. We appreciate the detailed feedback regarding experimental diversity and model sensitivity. Below, we address your concerns regarding data impact, rationale noise, and the specific trade-offs between our model variants, as well as other questions and comments.
>
> ### **\[W-1\] Impact of Number of Preference Samples**
> The reviewer raised a crucial question regarding the impact of preference sample size. To address this, we conducted an extensive scaling experiment on the 2-task ManiSkill setup, varying the dataset size from 200 to 2000 queries per task. Note that we keep the original experimental result run with 1000 queries.
>
> The results in Table 1 highlight a fundamental difference in how these methods behave as data increases. While baseline methods like BT-Multi and RFP show marginal improvements as data increases, they quickly hit a performance ceiling. For instance, on the Pick (OOD) task, BT-Multi hovers between 0.60 and 0.69. This occurs because these baselines maximize the likelihood of the confounded data; adding more samples merely reinforces their reliance on the spurious feature (e.g., object color) rather than correcting it.
>
> In contrast, ReCouPLe-EC successfully leverages the additional data. By anchoring the reward function to the provided rationale, the model uses the extra samples to sharpen the distinction between the causal feature and the confounders. Consequently, performance steadily improves with dataset size, rising from 0.733 $\rightarrow$ 0.913 on Pick (OOD) and 0.747 $\rightarrow$ 0.920 on Place (OOD).
>
> *Table 1: Scaling Analysis on ManiSkill (Reward Accuracy on 2-Task ManiSkill OOD)*
> | Preference Queries | 200 | 500 | 1000 (Original Result) | 2000 |
> | :---- | :---- | :---- | :---- | :---- |
> | *Task: Pick (OOD)* | | | | |
> | BT-Multi | 0.573 | 0.627 | 0.600 | 0.693 |
> | RFP | 0.540 | 0.613 | 0.620 | 0.760 |
> | **ReCouPLe-EC** | **0.733** | **0.820** | **0.820** | **0.913** |
> | **ReCouPLe-IC** | 0.547 | 0.580  | 0.633 | 0.793 |
> | *Task: Place (OOD)* | | | | |
> | BT-Multi | 0.667 | 0.760 | 0.820 | 0.767 |
> | RFP | 0.733 | 0.720 | 0.800 | 0.760 |
> | **ReCouPLe-EC** | 0.747 | **0.847** | **0.940** | **0.920** |
> | **ReCouPLe-IC** | **0.773** | 0.793 | 0.807 | 0.807 |
>
> ### **\[W-1\] Impact of Noisy Rationales and Diversity**
> The reviewer asked about the impact of "noisy" rationales and noted the lack of diversity in the original ManiSkill experiments. We address this both theoretically and empirically:
>
> * **Theoretical Formulation:** As discussed the ***Data generation*** in section 5.2 of the paper and the response to the reviewer **NmL7**, we inherently model noise by computing component-wise advantages and converting them to a softmax distribution from which we sample the reason behind the preference. This is grounded in Discrete Choice Theory \[1, 2\], which predicates that human decision-making is inherently stochastic. Humans do not simply choose the maximum, but the probability of selecting an option (*or citing a reason*) is proportional to its underlying utility or saliency. Similarly, in our original Meta-World experiments (Appendix B.3), we sampled reason texts via a **softmax distribution** rather than an **argmax**. This accounts for the stochasticity of human feedback \[3\], where users may cite valid but secondary factors.
> * **Empirical Robustness:** While our Meta-World experiments were designed with multiple underlying reasons the reviewer suggested, we acknowledge that our original ManiSkill experiments used a fixed, single-sentence reason that clarifies the true causal feature of *object size*. To verify that our method is not reliant on this rigidity, we conducted a new experiment using **16 distinct paraphrases** per task (e.g., *"cube is bigger"*, *"larger cube is picked"*, *"smaller object is not picked"*).
>
> *Table 2: Robustness to Rationale Diversity (Reward Accuracy on on ManiSkill OOD)*
> | Model | Pick (OOD) | Place (OOD) |
> | :---- | :---- | :---- |
> | BT (Baseline) | 0.540 | 0.830 |
> | **ReCouPLe-EC (Original)** | 0.820 | 0.940 |
> | **ReCouPLe-EC (Diversified)** | **0.867** | **0.967** |
> | **ReCouPLe-IC (Original)** | 0.633 | 0.807 |
> | **ReCouPLe-IC (Diversified)** | 0.620 | 0.820 |
>
> **Results:** As shown in Table 2, performance is **on-par and even improved** when using diversified reasons. Notably, **ReCouPLe-EC** improved from **0.820** to **0.867** on the Pick task. This confirms that ReCouPLe is robust to the semantic noise and phrasing diversity inherent in natural language. The results suggest that our trajectory encoder successfully learns robust representations, extracting the common causal features shared across varying sentence structures rather than overfitting to specific phrasing.
>
> \[1\] McFadden, Conditional Logit Analysis of Qualitative Choice Behavior, 1973
> \[2\] Luce, Individual Choice Behavior: A Theoretical Analysis, 1959
> \[3\] Yang et al., Learning from Comparative Language Feedback, CoRL 2024

---

> > ### Author Response · Authors · 2025-12-03
> > **Response to Reviewer fFvS (2/3)**
> >
> > ### **\[W-2\] Scalability (Sparse Explanations)**
> >
> > To address the concern that collecting rationales is expensive, we trained ReCouPLe where rationales were provided for only a subset (25% and 50%) of the preference pairs.
> >
> > *Table 3: Scalability to Sparse Explanations (Reward Accuracy on ManiSkill OOD)*
> >
> > | Model | Pick (OOD) | Place (OOD) |
> > | :---- | :---- | :---- |
> > | BT (Baseline) | 0.540 | 0.830 |
> > | **ReCouPLe-EC (25% Reasons)** | **0.783** | **0.940** |
> > | **ReCouPLe-EC (50% Reasons)** | 0.767 | 0.947 |
> > | **ReCouPLe-IC (25% Reasons)** | 0.673 | 0.807 |
> > | **ReCouPLe-IC (50% Reasons)** | 0.660 | 0.847 |
> >
> > **Results:** Even when **75% of reasons are missing**, ReCouPLe-EC retains high robustness (0.783) compared to the baseline (0.540).
> >
> > ### **\[W-4, W-3\] Generalization, Visual Causal Confusion, and Sensitivity Analysis**
> >
> > The reviewer expressed concern about performance sensitivity and requested more challenging test tasks. To address this, we designed an additional **Meta-World** experiment explicitly to highlight visual causal confusion under the presence of diverse reasons shared across different types of manipulation tasks.
> >
> > **Experimental Design:** We utilized the **Push** and **Push-Wall** tasks, introducing **background color** as a non-causal distractor feature.
> >
> > * **Confounded Data:** For *Push*, preferred trajectories were collected in environments with a **blue** background, while non-preferred trajectories had a **gray** background. For *Push-Wall*, this color scheme was flipped.
> > * **OOD Evaluation:** We tested the models in an environment where these background colors were swapped.
> > * **Diverse Rationales:** We maintained diverse reasons for preference (4 reasons for Push, 5 for Push-Wall), sharing high-level semantics (as in Appendix B.3). Crucially, we sampled these reasons from a **softmax distribution** of component-wise advantages, identical to the data generation process in Section 5.2.
> >
> > *Table 4: Visual Causal Confusion (Reward Accuracy on Meta-World OOD)*
> >
> > | Model | Push (Visual OOD) | Push-Wall (Visual OOD) |
> > | :---- | :---- | :---- |
> > | BT (Baseline) | 0.233 | 0.113 |
> > | BT-Multi | 0.367 | 0.227 |
> > | **ReCouPLe-EC** | 0.713 | 0.760 |
> > | **ReCouPLe-IC** | **0.793** | **0.900** |
> >
> > **\[W-3\] Performance Sensitivity (EC vs. IC Analysis)**:
> > The reviewer noted that performance varies between the EC and IC variants (e.g., EC is better in Table 1, IC is better in Table 4). We clarify that this is not fragility, but a result of the different mathematical properties suited for different environments. Each variant is more effective depending on the diversity of valid reasons and the complexity of the feature space in the environment, as detailed below:
> >
> > 1. **ReCouPLe-EC (Equality Constraint):** Enforces $r\_{\perp}(\tau\_A) \approx r\_{\perp}(\tau\_B)$. This assumes the rationale explains the *entire* preference difference. This strictness is optimal for **ManiSkill**, where the task is precise and the reason (e.g., "pick larger") is the dominant factor. Hence, EC outperforms IC in ManiSkill (Tables 1-3).
> > 2. **ReCouPLe-IC (Inequality Constraint):** Enforces $r\_{\parallel} \> r\_{\perp}$. This is a "softer" constraint that allows the orthogonal component to vary, provided it is less influential than the causal reason. This is superior in **Meta-World** (Table 4), where the environment has multiple features (with diverse corresponding reasons behind preference) related to the task success and is visually confounded. IC allows the model to suppress the confounder (background color) without over-constraining the difference in residual features necessary for capturing behaviors for robot control not mentioned in reasons.

---

> > > ### Author Response · Authors · 2025-12-03
> > > **Response to Reviewer fFvS (3/3)**
> > >
> > > ### **\[W-5, Q-2\] Correlation between Reward Accuracy and Policy Performance**
> > >
> > > We thank the reviewer for raising this insightful point. We agree that in general RLHF and PbRL settings, higher reward model accuracy does not always translate to better policies. This phenomenon is often attributed to reward overoptimization or "Goodhart's Law" effects \[4\], where models fit noise or non-robust features that policies subsequently exploit. Specifically, in \[4\], authors define **"Causal Goodhart"** as a scenario where a proxy reward model relies on features (e.g., answer length or background color) that correlate with the gold label but do not cause it. Standard reward models maximize these spurious correlations, leading to high training accuracy but poor policy performance when the agent exploits the non-causal feature. This has been also analyzed in the context of robotic policy learning \[5\].
> > >
> > > **ReCouPLe is explicitly designed to address Causal Goodhart.** By using the rationale to project the reward onto the causal axis ($r\_{\parallel}$), we force the model to ignore the spurious "Goodhart" features ($r\_{\perp}$). Consequently, in our setting, OOD reward accuracy becomes a **reliable binary detector of causal disentanglement**. A reward model that fails to disentangle the causal feature (e.g., relying on background color) will necessarily yield low accuracy on OOD validation pairs and fail to guide the policy.
> > >
> > > Our empirical results also confirm this: we observed a **Pearson correlation of 0.924** and a **Mean Spearman’s Rank correlation of 0.933** between OOD reward accuracy and policy success in MetaWorld. This suggests that in settings with strong spurious correlations, the reward model's OOD classification performance is a reliable proxy for its utility in policy training.
> > >
> > > ### **Additional Clarifications**
> > >
> > > * **\[W-5\] Comparison to True Reward:** The reviewer suggested comparing policy performance against a policy trained on the "true reward." In our experiments, we provided Behavioral Cloning (BC) results on preferred demonstrations as a proxy for "gold standard" behavior. We prioritized comparing against preference-based baselines to maintain a fair evaluation of reward learning methods.
> > >   **\[Q-1\] Dataset Size:** As detailed in **Appendix A.2**, we used 1,000 synthetic preference pairs per task for the ManiSkill state-based experiments and 500 for the image-based experiments.
> > > * **\[Q-4\] Learnable Parameters:** ReCouPLe uses the **same number of learnable parameters** as the multitask baselines. Across all methods, the language encoder (T5) is frozen and the only learnable component is the trajectory encoder. Thus, our performance gains stem from better geometric alignment in the embedding space, not increased model capacity.
> > > * **\[Q-3\] Performance Fluctuations (Table 1):** Regarding the specific instance where a baseline outperformed ReCouPLe (i.e., Pull task in Table 1 of the original manuscript), we attribute this to the stochasticity in the reward learning optimization. However, across all other experiments in the original manuscript and additional experiments, the aggregate trend strongly and consistently favors ReCouPLe, demonstrating its superior robustness and generalization.
> > >
> > > \[4\] Gao et al., Scaling Laws for Reward Model Overoptimization, ICML 2023
> > > \[5\] Tien et al. Causal Confusion and Reward Misidentification in Preference-based Reward Learning, ICLR 2023

---

### Official Review · Reviewer_mHRs · 2025-11-01

**Soundness:** 2
**Presentation:** 3
**Contribution:** 2
**Rating:** 4
**Confidence:** 4

**Summary:**

This proposes a method to reduce causal confusion in preference-based reinforcement learning, where reward models often overfit to spurious correlations. ReCouPLe augments standard binary preference data with short natural-language rationales, treating each rationale as a causal projection axis that aligns trajectory embeddings with the stated reasons. This decomposition isolates causal features that truly drive preferences while ignoring irrelevant correlations. Experiments on ManiSkill and Meta-World show that ReCouPLe significantly improves out-of-distribution robustness and enables zero-shot reward transfer to new tasks without additional annotations.

**Strengths:**

- Employs language-based causal alignment by combining simple preference data with natural-language rationales, guiding the model to focus on causally relevant features that reflect the user’s true intent.
- Demonstrates zero-shot reward transfer to unseen environments without requiring any additional preference collection or reward model training.

**Weaknesses:**

- The task instructions used are overly simple, and it is unclear whether incorporating language reasoning truly provides an advantage in this setup. It would strengthen the work to include experiments using more diverse language rationales or to analyze whether varying the linguistic expressions for the same rationale improves performance.
- The rationale extraction process appears heuristic and heavily dependent on ground-truth rewards (especially in MetaWorld). This reliance limits the method’s applicability to benchmarks where reward engineering is difficult or unavailable.
- Because each preference annotation requires a corresponding rationale, the method is not easily scalable to online RL settings. It would be valuable to discuss how ReCouPLe could be modified or extended for online applications.
- Prior work [1] has shown that image-based control tends to exacerbate causal confusion—particularly in MetaWorld—but the paper does not provide results for visual reward learning and policy training on MetaWorld to validate this.

**References**\
[1] Subtask-Aware Visual Reward Learning from Segmented Demonstrations, ICLR 2025.

**Questions:**

- It seems that the RFP baseline is just simple extension of Multi-BT operations twice, once for each of two distinct text instructions. Given this, what could explain RFP’s superior performance compared to Multi-BT, despite their apparent similarity?
- Could the authors clarify the specific domains or conditions under which ReCouPLe-EC and ReCouPLe-IC are most advantageous? At present, the explanations appear somewhat post-hoc, matched to results rather than grounded in theory.
- Can the method scale to setups with more than three distractors? For instance, beyond object size and color, could it handle additional factors such as object shape while maintaining consistent reasoning and performance?

---

> ### Author Response · Authors · 2025-12-03
> **Response to Reviewer mHRs (Part 1/3)**
>
> We thank the reviewer for their detailed evaluation and for recognizing the strengths of our method in leveraging language for causal alignment and zero-shot transfer. We appreciate the constructive feedback regarding linguistic diversity, scalability, and visual experiments. Below, we address your concerns with new experimental evidence and clarifications.
>
> ### **\[W-1\] Simplicity of Task Instructions & Linguistic Diversity**
>
> The reviewer noted that the original task instructions were simple and asked if language reasoning truly provides an advantage under diverse linguistic expressions. We agree that verifying robustness to phrasing is critical.
>
> * **Detailed Experimental Setup:** In the original setup, a single canonical reason was used. To verify linguistic robustness to phrasing and semantic diversity, we generated **16 distinct paraphrases** per task in our rebuttal experiments. These variations included:
>   * **Synonyms:** *"cube is bigger"*, *"object is larger"*
>   * **Specific Descriptors:** *"red cube is larger"*, *"blue object is bigger"*
>   * **Passive Voice:** *"larger cube is picked"*
>   * **Negation/Contrast:** *"smaller cube is not picked"*, *"tinier object is not placed"*
> * As shown in **Table 1**, training with these diverse rationales **improved** OOD performance even compared to the fixed-reason baseline (e.g., 0.867 vs 0.820). This demonstrates that ReCouPLe does not rely on simple pattern matching. The trajectory encoder is able to learn robust **semantic representations**, extracting the common causal features in the preferred trajectories and aligning them with language representations shared across diversified language reasons.
>
> *Table 1: Robustness to Rationale Diversity (Reward Accuracy on ManiSkill OOD)*
>
> | Model | Pick (OOD) | Place (OOD) |
> | :---- | :---- | :---- |
> | BT (Baseline) | 0.540 | 0.830 |
> | **ReCouPLe-EC (Original)** | 0.820 | 0.940 |
> | **ReCouPLe-EC (Diversified)** | **0.867** | **0.967** |
> | **ReCouPLe-IC (Original)** | 0.633 | 0.807 |
> | **ReCouPLe-IC (Diversified)** | 0.620 | 0.820 |
>
> ### **\[W-2\] Rationale Extraction & Reliance on Ground Truth**
>
> The reviewer expressed concern that our rationale extraction is heuristic.
>
> * We model the selection of reasons using a softmax distribution rather than a deterministic argmax to align with **Discrete Choice Theory**, specifically the Random Utility Model \[1\] and the Luce Choice Axiom \[2\]. These theories posit that human decision-making is inherently stochastic, where the probability of selecting an option (*or citing a reason*) is proportional to its underlying utility or saliency, rather than strictly maximizing it.
> * Similarly, we sample reason codes from a **softmax distribution** of component-wise advantages. This simulates a realistic, noisy human annotator who probabilistically cites relevant factors rather than always deterministically identifying the “best” reason that captures the difference between two trajectories. This is analogous to the formulation in recent works \[3, 4\] that samples comparative language feedback on behalf of preference feedback, with a softmax sampling from different feature values.
>
> \[1\] McFadden, Conditional Logit Analysis of Qualitative Choice Behavior, 1973
> \[2\] Luce, Individual Choice Behavior: A Theoretical Analysis, 1959
> \[3\] Yang et al., Learning from Comparative Language Feedback, CoRL 2024
> \[4\] Hirota et al., Active Reward Learning and Iterative Trajectory Improvement from Comparative Language Feedback, IJRR 2025

---

> > ### Author Response · Authors · 2025-12-03
> > **Response to Reviewer mHRs (Part 2/3)**
> >
> > ### **\[W-3\] Scalability & Robustness to Sparse Data and Different RL Settings**
> >
> > The reviewer noted that requiring a rationale for every preference pair limits scalability.
> >
> > * **Focus on Offline Learning:** Our primary focus is reward learning from **static offline preference datasets**. In this standard setting, we collect trajectories and the user indicates preference to pairs of trajectories prior to reward learning. Therefore, it is natural to recycle such trajectories for **Offline RL** by relabeling both preferred and non-preferred trajectories with the learned reward model, rather than requiring expensive real-time interaction.
> > * **Scalability Experiment (Sparse Explanations):** To directly address the concern about annotation cost and scalability, we investigated if our method works when rationales are sparse. That is, we design a case where some or most pairs of preferences are missing the underlying reason behind preferences, similar to standard PbRL. We trained ReCouPLe where rationales were provided for only a **subset** (25% and 50%) of the preference pairs. For pairs without reasons, we simply minimize binary cross-entropy loss on the reward function following the Bradley-Terry model.
> > * **Results (Table 2):** Even when **75% of reasons are missing** (25% availability), ReCouPLe-EC retains high robustness (0.783 accuracy on Pick OOD) compared to the baseline (0.540). This confirms that ReCouPLe is highly label-efficient; the causal signal from a small set of rationales propagates to the entire dataset.
> >
> > *Table 2: Scalability to Sparse Explanations (Reward Accuracy on ManiSkill OOD)*
> >
> > | Model | Pick (OOD) | Place (OOD) |
> > | :---- | :---- | :---- |
> > | BT (Baseline) | 0.540 | 0.830 |
> > | **ReCouPLe-EC (25% Reasons)** | **0.783** | **0.940** |
> > | **ReCouPLe-EC (50% Reasons)** | 0.767 | 0.947 |
> > | **ReCouPLe-IC (25% Reasons)** | 0.673 | 0.807 |
> > | **ReCouPLe-IC (50% Reasons)** | 0.660 | 0.847 |
> >
> > ### **\[W-4\] Visual Causal Confusion (Meta-World)**
> >
> > The reviewer correctly pointed out that image-based control exacerbates causal confusion and requested visual experiments. We agree with observations in prior work \[5, 6\] that visual control is particularly susceptible to spurious correlations.
> >
> > * **New Experiment:** We designed a **Visual Causal Confusion** setup in Meta-World using the **Push** and **Push-Wall** tasks, utilizing a **DrQ-style encoder** for pixel-based learning.
> > * **Confounder:** We introduced **background color** as a spurious feature. Within a given trajectory pair, a preferred trajectory is collected in a setting with a specific background color (e.g., Blue for *Push*), while a non-preferred one has another (e.g., Gray). We then evaluated on OOD environments where these colors were swapped.
> >
> > *Table 3: Visual Causal Confusion (Reward Accuracy on Meta-World OOD)*
> >
> > | Model | Push (Visual OOD) | Push-Wall (Visual OOD) |
> > | :---- | :---- | :---- |
> > | BT (Baseline) | 0.233 | 0.113 |
> > | BT-Multi | 0.367 | 0.227 |
> > | **ReCouPLe-EC** | 0.713 | 0.760 |
> > | **ReCouPLe-IC** | **0.793** | **0.900** |
> >
> > * **Results:** Baselines suffer from causal confusion by overfitting to the background color. Their reward models rely on this spurious feature and fail to identify correct behavior for task success. On the other hand, ReCouPLe successfully disentangled the visual confounder, maintaining high performance. This validates the findings of *Park et al. (2021)* [5] and *Kim et al. (2025)* [6] regarding the severity of visual confusion, and demonstrates ReCouPLe's ability to solve it.
> >
> > \[5\] Park et al., Object-aware regularization for addressing causal confusion in imitation learning, NeurIPS 2021
> > \[6\] Kim et al., Subtask-Aware Visual Reward Learning from Segmented Demonstrations, ICLR 2025\.

---

> > > ### Author Response · Authors · 2025-12-03
> > > **Response to Reviewer mHRs (3/3)**
> > >
> > > ### **Clarification on Questions**
> > >
> > > * **\[Q-1\] Why RFP outperforms Multi-BT:** RFP still takes advantage of common causal features mentioned in the reason. Specifically, RFP includes a reason score which encourages the trajectory encoder to align the common feature in preferred trajectories with the common reason behind preference across different tasks. However, RFP lacks **structural safeguards**: it has no explicit neutrality constraint ($r\_\perp$) to prevent the residual embedding from collapsing or leaking spurious correlations. ReCouPLe’s geometric projection strictly enforces this separation by orthogonally decomposing the trajectory embedding to reason-aligned and reason-orthogonal components.
> > > * **\[Q-2\]** The reviewer asked to clarify the conditions under which each variant is advantageous. We clarify that the performance difference is not fragility, but a predictable result of how the constraints interact with environment complexity:
> > >   * ReCouPLe-EC (Equality Constraint): Enforces $r\_{\perp}(\tau\_A) \approx r\_{\perp}(\tau\_B)$. This imposes a strict condition that the entire preference difference must be explained by the rationale. This strictness is optimal for domains like ManiSkill, where the task is precise and the stated reason (e.g., "pick larger") is the single dominant factor driving the preference. Consequently, EC consistently outperforms IC in our ManiSkill experiments (Tables 1-2)
> > >   * ReCouPLe-IC (Inequality Constraint): Enforces $r\_{\parallel} \> r\_{\perp}$. This is a "softer" constraint that allows the orthogonal component to vary, provided it is less influential than the causal reason. This proves superior in Meta-World (Table 3), where there are multiple possible reasons behind preference corresponding to different feature components in the ground-truth reward of each task. ReCouPLe-IC accommodates scenarios where a preferred trajectory excels in the stated reason but may be suboptimal in other unstated, secondary aspects. The Inequality Constraint also allows the model to suppress the strong confounder (e.g., background color) without over-constraining the difference between two trajectories regarding residual trajectory features necessary for basic control.
> > > * **\[Q-3\] Scaling to \>3 Distractors:** Our new Meta-World visual experiment effectively models this high-dimensional scenario. The environment contains **multiple causal features** (distance to object, distance to goal) and a non-causal, visual confounder (background). ReCouPLe-IC's strong performance (0.900) confirms it can handle complex, multi-factor distractor settings.

---

### Meta-Review · Area_Chair_MLJe · 2026-01-07

**Summary:**

All reviewers except one voted to reject this paper. On one hand they appreciated the zero-shot results and the clarity of the paper. On the other they had issues with missing experiments. The authors were able to respond to each of these concerns convincingly, inventing experiments when the reviewer concern was underspecified. For this reason, I vote to accept.

**Reviewer Concerns:**

Please see above.

**Reviewer Scores:**

I believe the reviewers would have kept their scores or increased them.

---

### Decision · Program_Chairs · 2026-01-26

Accept (Poster)